# Integrated Framework to Assess the Extent of the Pandemic Impact on the Size and Structure of the E-Commerce Retail Sales Sector and Forecast Retail Trade E-Commerce

Cristiana Tudor 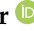

International Business and Economics Department, Bucharest University of Economic Studies, 010374 Bucharest, Romania; cristiana.tudor@net.ase.ro

**Abstract:** With customers' increasing reliance on e-commerce and multimedia content after the outbreak of COVID-19, it has become crucial for companies to digitize their business methods and models. Consequently, COVID-19 has highlighted the prominence of e-commerce and new business models while disrupting conventional business activities. Hence, assessing and forecasting e-commerce growth is currently paramount for e-market planners, market players, and policymakers alike. This study sources data for the global e-commerce market leader, the US, and proposes an integrated framework that encompasses automated algorithms able to estimate six statistical and machine-learning univariate methods in order to accomplish two main tasks: (i) to produce accurate forecasts for e-commerce retail sales (e-sale) and the share of e-commerce in total retail sales (e-share); and (ii) to assess in quantitative terms the pandemic impact on the size and structure of the e-commerce retail sales sector. The results confirm that COVID-19 has significantly impacted the trend and structure of the US retail sales sector, producing cumulative excess (or abnormal) retail e-sales of $227.820 billion and a cumulative additional e-share of 10.61 percent. Additionally, estimations indicate a continuation of the increasing trend, with point estimates of $378.691 billion for US e-commerce retail sales that are projected to account for 16.72 percent of total US retail sales by the end of 2025. Nonetheless, the current findings also document that the growth of e-commerce is not a consequence of the COVID-19 crisis, but that the pandemic has accelerated the evolution of the e-commerce sector by at least five years. Overall, the study concludes that the shift towards e-commerce is permanent and, thus, governments (especially in developing countries) should prioritize policies aimed at harnessing e-commerce for sustainable development. Furthermore, in light of the research findings, digital transformation should constitute a top management priority for retail businesses.

**Keywords:** e-commerce; automated forecasting; COVID-19 impact; TBATS; neural networks

## 1. Introduction

The COVID-19 pandemic has been labeled a black swan event [1] with far-reaching socio-economic consequences [2–4]. At the world level, 200 countries and territories have reported COVID-19 cases [5]. Consequently, closing substantial segments of the physical economy has promoted digitization by enhancing the use of digital channels, particularly e-commerce [6–9]. As such, unlike any other time in history, the e-commerce sector played a key role in society, allowing consumers to safely acquire goods through strict regulations aimed at safeguarding public health [10]. Consequently, in the context of a global economic downturn, e-commerce has surged while digital transformations have accelerated [11,12]. Several nations have experienced shifts towards e-commerce, particularly along the food supply chain. Examples include farmers who began using digital technologies to sell their produce directly to consumers or restaurants that began offering food or grocery delivery services [13]. Moreover, the pandemic has produced an excess consumer interest and subsequent growth in the global web and videoconferencing Software as a Service (SaaS)

market that would not have occurred in the absence of the pandemic [14]. Concomitantly, the increase in digitization caused a disruption in traditional business methods such as supply chain management and distribution [15]. Companies that formerly focused on selling through offline channels expedited their digital transformation in order to respond to the fast-evolving commercial reality ([10,16,17]). Hence, the resulting shift from brick-and-mortar retail to e-commerce has spurred total online retail sales and increased the share of e-commerce in total retail sales ([13,18,19]), while total retail sales have decreased. Global e-commerce sales, including both business-to-business (B2B) and business-to-consumer (B2C) sales, increased to $26.7 trillion in 2019, up 4% from 2018, making for 30% of the 2020′s global gross domestic product (GDP) [20]. Additionally, global e-retail sales, comprising B2C sales, equaled $4.2 trillion in 2021 [21] and are expected to reach $5.42 trillion in 2022 and $6.38 trillion by 2024 [22]. The United States (US) leads the global e-commerce market, followed by Japan and China, whereas China, the US, and the United Kingdom are the top three countries in terms of B2C e-commerce sales [23].

Concurrently, accurate sales forecasting is of paramount importance in retailing, as retail businesses rely on sales predictions for various operational decisions [24], including decisions regarding planning production, purchasing, transportation, and workforce [25]. Accurate projections are especially critical for retail and consumer-oriented industries, namely the electronic market and the fashion business [26]. Furthermore, retail sales have long been recognized as a leading or coincident economic indicator [27–30], whereas e-commerce retail sales have been shown to have an even stronger impact on regional development [31]. Moreover, e-commerce has a documented mitigating effect on pollution, with estimates indicating that offline shopping produces between 1.5 and 2.9 times more greenhouse gas emissions than online shopping [32]. Thus, unsurprisingly, the e-commerce sector is regarded as a key driver for the circular economy [10] and robust estimates of its future evolution are key for effective and efficient policymaking.

Additionally, time series prediction is acknowledged as an important technology to promote industrial intelligence [33]. Consequently, e-market planners and other planning agencies in the information technology sector are increasingly recognizing the importance of assessing and forecasting e-commerce growth [34].

In light of the above considerations, a quantitative assessment of the impact of the pandemic on the e-commerce sector (and in particular its persistence) has emerged as timely research topic, with important economic and policy implications that have in turn motivated the current study. Considering previous findings that confirm the role of retail sales as an economic leading indicator, the univariate modeling approach, where data speak for themselves [35,36], stands as the most appropriate forecasting approach.

Three aspects define this paper's primary contributions: (1) it develops an integrated framework for forecasting total e-commerce retail sales and the share of e-commerce in total retail sales that embeds an array of relevant statistical and machine-learning forecasting methods; (2) it assesses empirically the magnitude of the COVID-19 pandemic's impact on the size and structure of the e-commerce retail sales sector; and (3) it investigates the short and long-run effects of the pandemic on this sector and thus explores whether the change in consumer behavior induced by the COVID-19 pandemic towards e-commerce is persistent.

To answer these research goals, long-term forecasts for total US e-commerce retail sales and the share of e-commerce in total US retail sales are generated by the model that outperforms in terms of out-of-sample predicting performance embedded in the framework. Results show that TBATS (exponential smoothing state space model with Box-Cox transformation, ARMA errors, trend, and seasonal components) can better capture the specificities of the e-sales time series, whereas both TBATS and STS (structural time series Model) show a similar superior ability for explaining and predicting e-share trends. Furthermore, new indicators reflecting excess (or abnormal) e-commerce retail sales (i.e., excess e-sales) and excess (or abnormal) share of e-commerce in total retail sales (i.e., excess e-share) are proposed and estimated through the integrated framework by the

over-performing models. The findings indicate that the pandemic has accelerated the growth of the e-commerce sector in size and has expanded, above expectations, its weight in total retail sales. Consequently, the pandemic induced a total of $227.820 billion in excess e-sales and has increased the weight of e-commerce in total retail by 10.61 percent, thus accelerating its structural change. In the absence of the pandemic, these evolutions would not have been observed before 2026. Our estimations thus show that the unseen black swan event accelerated the evolution of the e-commerce sector by at least five years. Furthermore, research findings confirm that the COVID-19 pandemic will modify consumer choices in the long run toward e-commerce, as estimations indicate overall US e-commerce sales to increase through the end of 2025, reaching $378.691 billion in absolute terms, and corresponding to a share in the US retail sales sector of 16.72 percent by December 2025.

The remainder of the study is organized as follows. Section 2 provides a short overview of the related literature. Section 3 firstly covers the data and discusses its historical trends, then explains the method and presents the integrated framework proposed in this study, along with its main building blocks. Next, Section 4 presents the empirical results and checks their robustness. Section 5 proceeds with a discussion of findings, while Section 6 concludes the study and extracts some policy implications.

## 2. Literature Review

Producing accurate forecasting systems for real-world time series modeling is a challenging task [37]. Previous studies that attempt to forecast retail sales are based on forecasting methods that belong to one (or both) of the two main categories (i.e., "cultures") delineated by [38], each with its strengths and weaknesses [39]: econometrics (i.e., statistical methodologies) and machine learning (self-learning systems that learn from past data to enhance their predictive performance) [40]. They are united by the common goals of providing, on the one hand, information about the data-generating process and, on the other hand, predictability for its future evolution [41]. Machine-learning techniques have the important advantage of being data-driven and needing minimal a priori assertions about the data-generating process [42–44] while being capable of capturing non-linearity that characterizes many empirical time series. This important characteristic contributed to making machine-learning techniques such as artificial neural networks (ANNs) very useful in many circumstances where no theoretical direction for an acceptable data generation process is available, and thus the default method of choice for modeling real-world phenomena [45,46].

However, both in practice and in research, forecasting models are usually chosen manually to account for various data-specific elements, such as autocorrelation, trend, seasonality, or exogenous variables [47]. In particular, retail sales series have been shown to typically contain trend and seasonal patterns, and thus pose increased difficulties in the development of reliable forecasting systems [25]. Given the particularities of retail sale data, previous studies tend to disagree in identifying the best-performing forecasting model, which further complicates the task. Some authors have found that statistical models are able to provide accurate forecasts of retail sales (among others, [25,48,49]) whereas others have identified machine-learning methods to out-perform ([50–55]). Thus, previous research endeavors have been characterized by mixed results in terms of the effectiveness of statistical methods compared with machine-learning methods to model and predict retail sales.

This study contributes to extending this strand of literature with new and robust findings based on the most recent available data. Additionally, it goes further than previous research by quantitatively assessing the impact of the COVID-19 pandemic on the e-commerce trend and structure and by providing reliable estimates for the future evolution of the sector, thus answering current open questions regarding the magnitude and persistence of this significant impact. As such, this study brings new evidence on the extent to which the pandemic has impacted the US e-commerce retail sales sector and explores whether these disruptions are reversible or persistent. In the process, it develops a

forecasting instrument (i.e., the integrated framework), which is capable of improving the forecasting accuracy of e-commerce series.

## 3. Materials and Methods

### 3.1. Data

The research conducted in this paper employs quarterly data for the two main variables of interest (i.e., e-commerce retail sales [ECOMSA] and the share of e-commerce in total retail sales [ECOMPCTSA]) sourced from the Federal Reserve Bank of St. Louis's (FRED) database, which gathers data from the U.S. Census Bureau. The sample period extends from October 1999 (i.e., 1999 Q3) to October 2021 (i.e., 2021 Q3) and therefore includes 88 quarterly observations for each e-commerce time series. The timeframe provides a sufficing sample that includes both tranquil periods, as well as the recent black swan event (i.e., the COVID-19 pandemic), through which the forecasting ability of a pool of predictive models (statistical and machine-learning) encompassed into an integrated forecasting framework can be tested and indicators of excess e-commerce retail sales can be estimated.

For a quick overview of historical evolutions in the retail sales sector, which also includes a sectorial perspective, we also make use of relevant data retrieved from the US Census Bureau database. As such, sector-level time series for Annual U.S. Retail Trade Sales (Total and E-commerce) covering the 1998–2015 period are sourced from Census for providing a deeper insight into historical trends and structure of the retail sector in the United States. This in turn contributes to identifying the most appropriate forecasting method for estimating excess e-sales/e-share that will be discussed further. To our knowledge, our analysis makes use of the most recent and complete available e-commerce retail sector data set for the leading global e-commerce market (i.e., the US market).

It is worth noticing that the US e-commerce retail sector had registered significant growth over the last decade, showing a more accelerated increase than total retail sales in the pre-pandemic era (Figure 1, panels a and b).

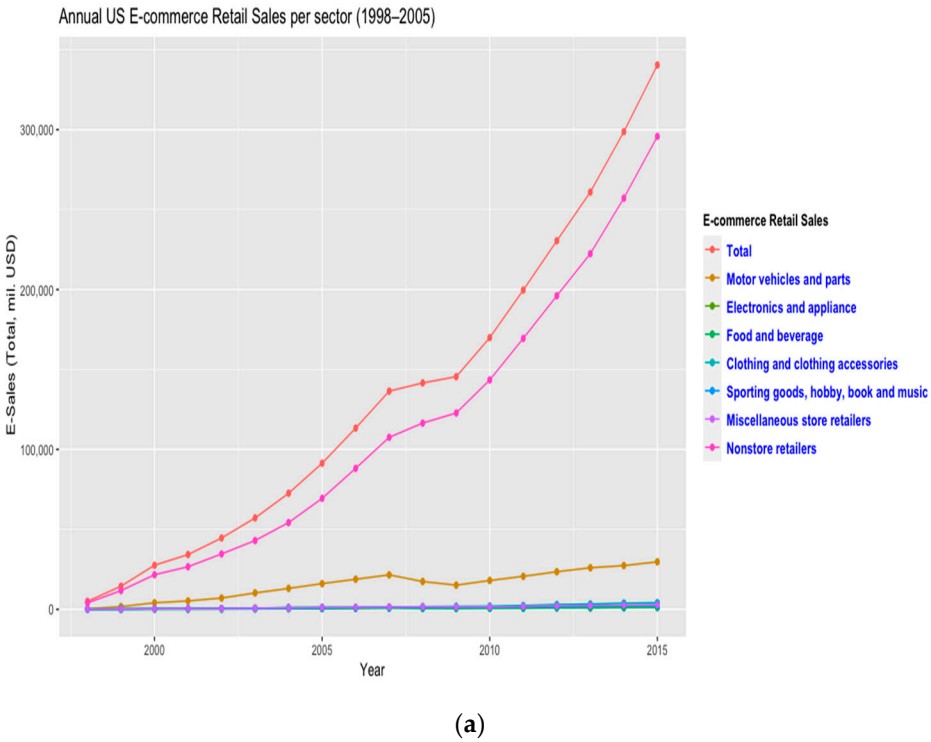

**(a)**

**Figure 1.** *Cont.*

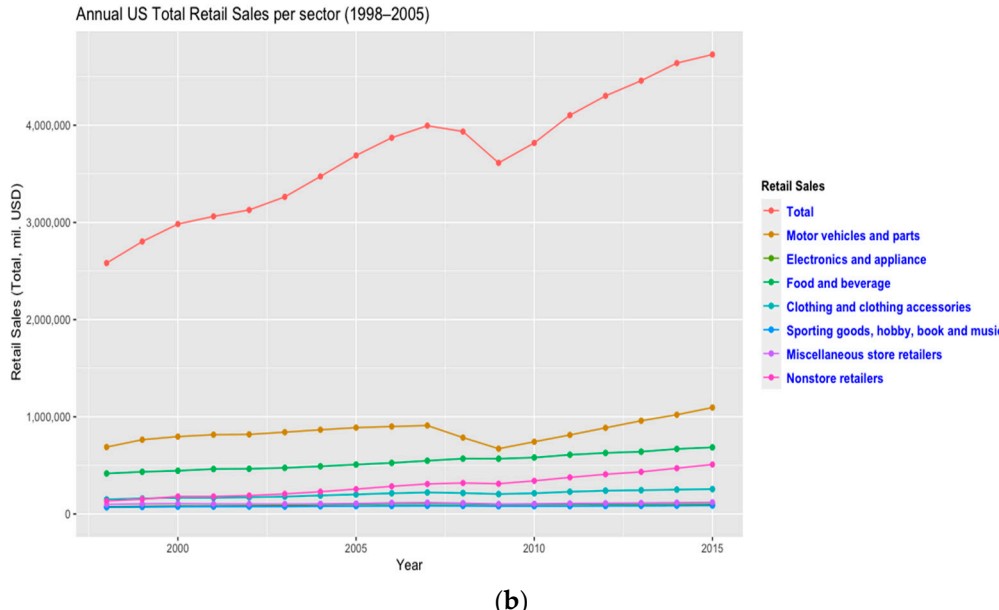

(**b**)

**Figure 1.** Historical trends and structure of US e-commerce retail sales per sector (panel **a**); historical trends and structure of US total retail sales per sector (panel **b**).

The driving engines behind retail sales growth have also been different. Thus, a sectorial perspective on the US market provided in Figure 1 reveals that the growth in total retail sales has been mainly driven by the motor vehicles and parts sector, whereas non-store retailers have been the driving force behind e-commerce retail sales.

However, visualization of the updated e-commerce series confirms that historical evolutions have been significantly disrupted by the COVID-19 pandemic [56]. Figures 2 and 3 show that e-commerce retail sales and the share of e-commerce in total retail sales reached an initial peak in the second quarter of 2020 and, after a mild reversal, accelerated again from January 2021 and continued through the year.

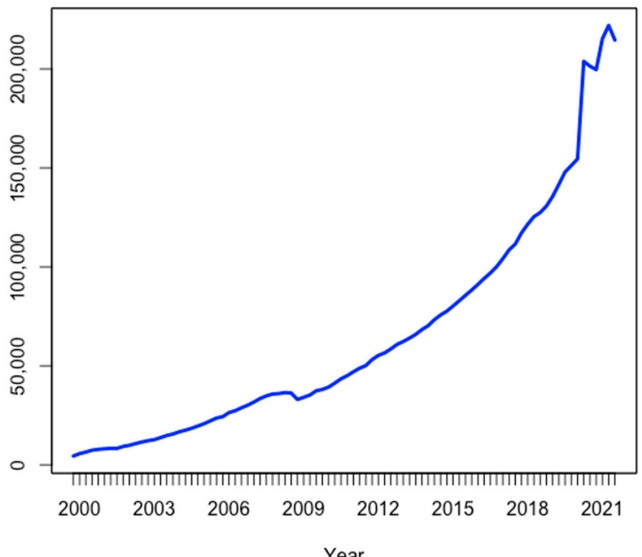

**Figure 2.** Quarterly evolution of E-Commerce Retail Sales [ECOMSA] over 1999–2021; Author's representation with data retrieved from the Federal Reserve Bank of St. Louis (FRED), which sources data from U.S. Census Bureau; https://fred.stlouisfed.org/series/ECOMSA; accessed on 8 September 2022.

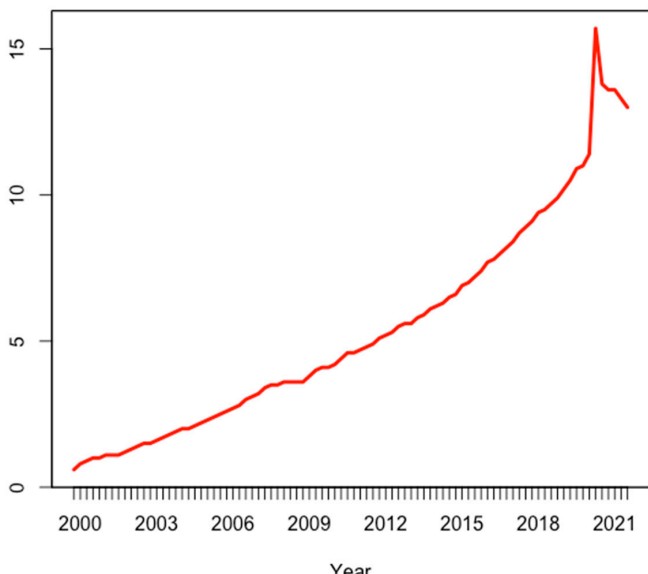

**Figure 3.** Quarterly evolution of the share of E-Commerce in Total Retail Sales [ECOMPCTSA] over 1999–2021, Author's representation with data retrieved from the Federal Reserve Bank of St. Louis (FRED), which sources data from U.S. Census Bureau; https://fred.stlouisfed.org/series/ECOMPCTSA, accessed on 8 September 2022.

The share of e-commerce in total retail sales followed a similar trend after the pandemic outbreak, registering a significant increasing trend over the first half of 2020. However, the two time-series diverge during 2021, when the share of e-commerce continued its downward trend that started in June 2020, whereas e-commerce retail sales reversed it. Hence, the share of e-commerce in total U.S. retail sales equals 13 percent in the third quarter of 2021, down from its peak level of 15.7 percent in the second quarter of 2020. On the other hand, retail e-commerce sales surpassed $214 billion in October 2021, whereas its highest historical level was registered in the previous quarter, reaching an all-time high value of $221.9 billion in the second quarter of 2021.

Overall, as of October 2021, e-commerce accounts for 13 percent of total retail sales in the United States, up from 0.6 percent in December 1999, whereas physical retail still accounts for 87 percent at the end of the analysis period. From 1999 Q4 to 2021 Q3, e-commerce has increased by $210.11 billion to a total of $214.586 billion, which translates into an impressive 4694% growth. Over the pandemic period, the growth of e-commerce has significantly accelerated. Relative to its 2019 Q4 level of $135.629 billion, e-commerce grew by 58.21%, and its penetration increased by 2%.

These historical trends, and especially the increased volatility induced by the COVID-19 pandemic, indicate that a single model may be unable of capturing efficiently all characteristics, evolutions, or available information hidden in the series and consequently can produce unreliable forecasts [57]. This in turn leads us to use a variety of statistical and machine-learning methods, both linear and nonlinear and analyze their performance from a comparative perspective. The best-performing model will then be used to produce forecasts of the variables of interest that will serve as proxies for the "business as usual" scenario for the e-commerce sector (i.e., without the black swan event that has disturbed its evolution).

*3.2. Method*

3.2.1. Integrated Framework

The integrated framework proposed in this research is reflected in Figure 4.

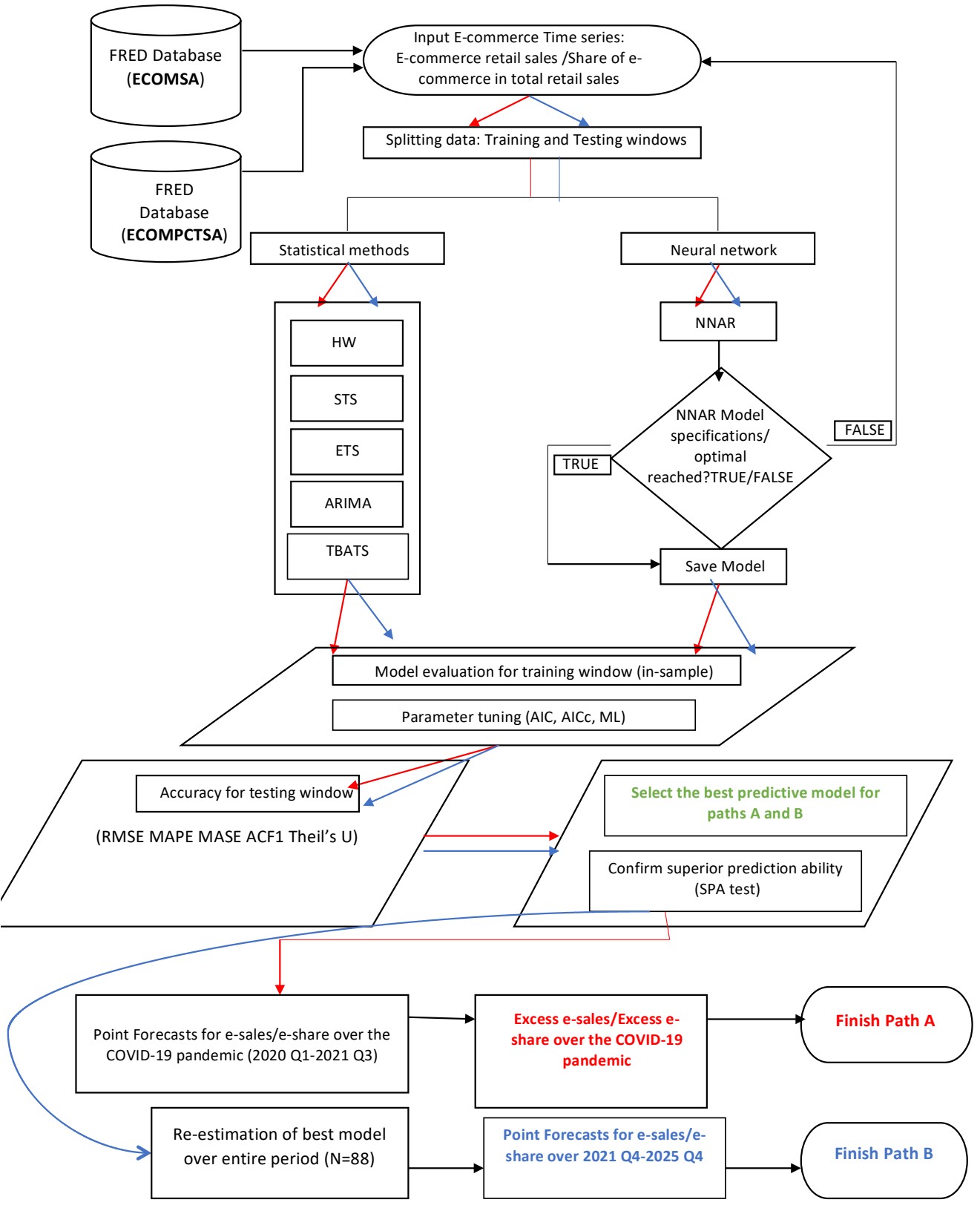

**Figure 4.** Integrated framework for forecasting e-commerce sector variables (i.e., e-commerce retail sales ECOMSA) and share of e-commerce in retail sales (ECOMPCTSA) where the algorithm follows subsequently path A (red color) and path B (blue color).

As depicted in Figure 4, the algorithm that is run within the integrated framework consecutively follows two distinct paths, which are delineated by different data splitting rules. Its main building blocks are presented in more detail in the following sub-sections.

The implementation of the integrated framework and all estimations are carried out in R software.

### 3.2.2. Data Splitting Rules and the Forecasting Technique

For forecasting purposes, each e-commerce time series of length $N = 88$ (from October (or Q3) 1999 to October (Q3) 2021) are separated into two subsets denoting a training period and a test period.

Firstly, to develop the first path (path A) followed by the algorithm on which the integrated framework depicted in Figure 4 is run, the data up to March (i.e., Q1) 2018 (representing 74 quarterly observations) are used in-sample for modeling and validation, and the data from 2018 Q2 to 2019 Q4 (i.e., 7 observations) are used to gauge how well the predictive models perform when forecasting beyond the sample. As the COVID-19 pandemic has significantly altered retail sales series trends and structure, we are especially interested in finding the best forecasting model over the test period or lead-time and subsequently using it for providing h steps ahead forecasts so as to cover the pandemic period ranging from January 2020 to October 2021 (i.e., 2020 Q1 to 2021 Q3).

Thus, in the first investigation (path A), the forecasting horizon $h$ is set to 7 (so as to cover the COVID-19 pandemic forecasting horizon that spreads over seven quarters), and point-forecasts for the seven quarters of the forecasting horizon are produced within the integrated framework by the over-performing model. As a consequence, we issue the most precise forecasts obtainable for the expected e-commerce retail sales and for the share of e-commerce in total retail sales (excess e-share) from the period spanning from January 2020 to October 2021 (over the COVID-19 pandemic period) based on quarterly historical data from January 1999 to December 2019, that is, in the absence of the COVID-19 pandemic. Real e-commerce sector data is subsequently compared to point forecasts, giving estimates of abnormal (excess) e-commerce retail sales and corresponding abnormal share of e-commerce in total retail sales e-sales.

In a subsequent investigation (path B followed by the algorithm), we take a different data splitting approach. Consequently, the training window spreads over the first 81 observations in the sample (i.e., 91 percent of observations, covering the period 1999 Q4–2019 Q4), and a testing period spanning from 2020 Q1- 2021 Q3 (thus a length of 7). For each e-commerce time series, point forecasts are produced after the six predictive models have been fit on the newly specified training window and their respective out-of-sample predictive ability has been analyzed on the new test set data. This approach ensures the robustness of results while also providing estimates for the future evolution of the two e-commerce time series based on the most recent available data. Hence, point forecasts for the e-commerce retail sales and for the share of e-commerce in total retail sales issued through the integrated framework and produced by the over-performing model for each time series until Q4 2025 bring further evidence on whether the impact of the COVID-19 pandemic on the e-commerce sector is long-lasting (otherwise, a revolving trend should be expected).

Hence, to automatically implement the method, the Algorithm 1 performs the following tasks running through Path A:

| **Algorithm 1**. Sequential steps for excess e-sales/e-share estimation |
|---|
| INPUT: Call Quandl() to source FRED data (ECOMSA, ECOMPCTSA series) |
| Step 1. Split each series: 74-7-7 rule (Train/Test/Forecast windows) |
| Step 2. Model fiting on ECOMSA and ECOMPCTSA training windows |
| 2a.call functions HoltWInters(),StructTS(), auto.arima(), ets(),tbats(),nnetar() |
| 2b.establish corresponding fitness function |
| 2c. determine best-fit model specifications for ECOMSA and ECOMPCTSA (2 x 6 best-fit models, in-sample setting) |
| 2d. check the residuals of the 2 $x$ 6 best-fit models from step 2c (ACF plots, PACF Plots, the Box-Ljung test); |
| IF (white noise), go to |
| Step 3. use step 2c models to produce out-of-sample forecasts over the test window for ECOMSA and ECOMPCTSA |
| ELSE, |
| repeat steps 2 a–d |
| Step 4. Compute performance metrics (MAE, MAPE, MASE, RMSE, Theil's U, ACF1) for the 2 $x$ 6 best-fit models |
| Step 5. Identify best out-of-sample predictive model for ECOMSA and ECOMPCTSA (2 $x$ 1 best forecasting model) |
| Step 6. Confirm superior forecasting ability for Step 5 models (call "DM.test") |
| Step 7. Re-estimate the Step 5 models on extended Train + Test window (i.e., pre-COVID-19 data, or 81 quarterly observations); retune model parameters; check residuals |
| Step 8. Use updated model specifications from Step 7 to produce forecasts for ECOMSA and ECOMPCTSA up to Q3 2021 |
| Step 8. Estimate and print excess e-sales/e-share |
| FINISH |

Specifically, the following script snippet (Box 1) clarifies the extraction of FRED data (i.e., total ecommerce retail sales) via Quandl and the implementation of the data splitting rule for path A:

**Box 1.** Script for data sourcing and splitting strategy for path A

```
library(Quandl)
Quandl.api_key(" . . . ") #personal Quandl key here
esales <- Quandl("FRED/ECOMSA",
                 type = "xts",
                 collapse = "quarterly",
                 start_date = "1999-10-01",
                 end_date = "2022-09-01")
x <- ts(esales)
test_x <- window(x, start=c(75,1),end=c(81,1))#testing window
x <- window(x, end=c(74,1)) #training window
```

Alternatively, for path B, the algorithm follows a similar, albeit not identical, sequence for each of the two e-commerce series. Thus, in step one it applies a different splitting rule (i.e., 71-17-17) and consequently, during Step 7, it re-estimates the over-performing forecasting model identified in Step 5 on an extended and updated window of data covering the train and test sets and thus comprising the entire time-series data of 88 quarterly observations spanning from 1999 Q4 to 2021 Q3. Finally, it produces predicted values for a forecasting horizon going up to the end of 2025 and hence it does not estimate excess statistics in this path.

*3.3. Excess (or Abnormal) E-Commerce Retail Sales (Excess E-Sales) and Excess Share of E-Commerce in Total Retail Sales (Excess E-Share)*

In this article, similar to the approach of the Corporate Finance Institute (CFI) [58] and of the European Commission ([59]), excess or abnormal e-sales refers to the reported

total value of retail trade e-commerce sales (or e-sales), above what could be anticipated under 'normal' conditions. We thus argue that this approach gives a general measure of the impact of the crisis on the e-commerce retail trade sales. The abnormal e-commerce sales statistic not only highlights the impact of the pandemic but also allows for further comparison of additional retail e-commerce sales induced by the pandemic among the world countries.

Excess (abnormal) e-sales statistic is thus expressed as the level of unanticipated e-sales in a month, compared with a "baseline" or benchmark level that represents expected e-sales over the time frame associated with the COVID-19 epidemic, in the absence of the pandemic, as in Equation (1):

$$y_h = E_{S_h} - \hat{E}_{S_h} \tag{1}$$

where $E_{S_h}$ is the observed level of e-sales at the forecasting horizon $h$ and $\hat{E}_{S_h}$ is the expected or baseline level of e-sales at $h$ under normal conditions. Although an alternative approach would be to use averages of last observations as a baseline level (see for example how the United Kingdom Office for National Statistics estimates excess deaths due to the pandemic), this method is less appropriate for data with trend or seasonality that characterizes retail sales series. Consequently, in this research the predicted level of e-sales at step $h$ issued with the optimal forecasting model is employed as the benchmark. We thus comparatively explore the performance of six commonly employed univariate time series predictive models and generate reliable point-estimates for expected e-sales levels from January 2020 to October 2021 based on quarterly historical data from December 1999 to December 2019, that is, in the absence of the COVID-19 pandemic. As such, these estimates that proxy the evolution of the e-sales series during "normal" conditions are compared to the real evolution of the series over the seven quarters that correspond to the calendar pandemic period, and abnormal (or excess) statistics are computed.

In a similar manner, excess e-share is estimated as the percentage rate of unexpected additional share of e-commerce in total retail sales:

$$y_{share_h} = E_{S\_share_h} - \hat{E}_{S\_share_h} \tag{2}$$

where $E_{S\_share_h}$ is the observed share of e-commerce in total retail sales (e-share) at the forecasting horizon $h$ and $\hat{E}_{S\_share_h}$ is the expected/benchmark level of e-share at $h$ under normal conditions.

The next subsection provides a background on the methods that are encompassed in the integrated framework for forecasting purposes.

### 3.4. Time Series Forecasting Models

We employ six univariate time series models that are embedded into the integrated framework to automatically model and predict two relevant e-commerce series: e-commerce retail sales and the share of e-commerce in total retail sales.

The HW (Holt-Winters) model developed by [60,61] is known as double exponential smoothing and is especially fit for trended and seasonal time series. As such, HW proved reliable for forecasting business data with seasonality, shifting trends, and seasonal correlation [62] and has thus been found to produce accurate short-term forecasts for sales or demand time-series data [63]. HW incorporates an exponential component ($E_t$) as well as a trend component ($T_t$), such that:

$$E_t = wY_t + (1-w)(E_{t-1} + T_{t-1}), 0 < w < 1 \tag{3}$$

and

$$T_t = v(E_t - E_{t-1}) + (1-v)T_{t-1}, 0 < v < 1 \tag{4}$$

After the exponentially smoothed and trend components, $E_t$ and $T_t$, are calculated for each observed value, the k-step-ahead forecast is produced by:

$$F_{t+k|t} = E_t + kT_t \tag{5}$$

The forecasting framework developed in this study automatically estimates the HW model by using the function "HoltWinters" in the "stats" package [64] where an additive" (the default) model is specified and nine start periods are used in the autodetection of start values. The function performs HW filtering for each time series that is fed to the framework (i.e., e-sales and e-share); then, by minimizing the squared prediction error, the algorithm automatically discovers the parameters for the best in-sample HW model.

The exponential smoothing state space model (ETS) has received extensive attention in the recent years since [65] proposed the state-space formulation as an extension of the exponential smoothing (ES) classical method. Exponential smoothing considers each time series as a combination of three components, namely, the trend (T), seasonal (S) and error (E) components, whereas the trend component is at its turn a combination of a level term (l) and a growth term (b). None (N), additive (A), additive damped (Ad), multiplicative (M), or multiplicative damped (Md) trend and seasonal components are all possible [66]. The final model is a three-word string (Z,Z,Z), where the first letter indicates the state space model's error assumption, the second the trend type, and the third the season type [67]. In this study, the integrated framework runs the "ets" function in R's "forecast" package [68,69] to automatically fit ETS models and thus to automatically select the error, type and season. Then, the corrected Akaike information criterion (AICc) is applied within the framework to find optimal ETS parameters for each time series.

Box and Jenkins [70] pioneered autoregressive integrated moving average (ARIMA) models [71], which are now one of the most widely used statistical methods for linear time series forecasting [72], prominently employed in economic time series forecasting [73]. ARIMA models are capable of capturing both nonseasonal and seasonal patterns of data, which might make them particularly useful for retail sales forecasting purposes [25]. Following the terminology in [74], a seasonal model is written as ARIMA *(p,q,d)(P,Q,D)s*, where s denotes the seasonal period, while the lowercase and the capital letters represent respectively the number of nonseasonal and seasonal parameters for each of its components. In equation form, an ARIMA *(p,d,q)(P,D,Q)s* model is written as:

$$(1 - \varphi_1 B - \ldots - \varphi_p B^p)(1 - \Phi_1 B^s - \ldots - \Phi_P B^{sP})(1 - B)^d (1 - B^s)^D Y_t = \\ (1 - \theta_1 B - \ldots - \theta_q B^q)(1 - \Theta_1 B^s - \ldots - \Theta_P B^{sQ})\varepsilon_t \tag{6}$$

where $\varepsilon_t$ is a random variable with mean zero and the standard deviation $\sigma$.

In this study, the integrated forecasting framework automatically trains ARIMA models with the "auto.arima function" (R "forecast" package). The function calculates unit root tests, along with reducing the AICc and MLE to identify parameters through a step-wise process, and determines whether the data used to train the model requires a seasonal differencing. Hence, the algorithm minimizes the corrected Akaike information criterion (AICc) as per Equation (7) to determine the values of $p$ and $q$, and then identifies the optimal ARIMA model that represents the smallest AICc within the estimated specifications. Thus, it provides a significantly increased efficacy as it chooses to navigate the model space using a step-wise search rather than taking into account every conceivable pairing of $p$ and $q$ [75].

$$AIC_c = AIC + \frac{2(p + q + k + 1)(p + q + k + 2)}{T - p - q - k - 2} \tag{7}$$

where $k = 1$ if $c/= 0$ and 0 otherwise, L is the maximum likelihood of the data and the last item in parentheses is the number of parameters in the model (including the variance of the residuals $\sigma^2$).

TBATS (the exponential smoothing state space model with Box-Cox transformation, ARMA errors, trend, and seasonal components) has been developed by [76]. Its main advantage lies in its capability to deal with complex seasonal behaviors of data [77]. The model representation is expressed as TBATS($\omega$, p, q, $\varphi$, {$m_1$, $k_1$}, {$m_2$, $k_2$}, ... ,{$m_T$, $k_T$}), where $\omega$ denotes the Box-Cox transformation, k is the quantity of harmonics used to represent the seasonal attribute, and $\varphi$ denotes the dampening parameter. The framework uses the "tbats" function from the "forecast" package to automatically fit TBATS models and choose the best parameters based on AIC.

The structural time series (STS) method [78] asserts a time-series is made up of a range of unobserved components, such as (deterministic and stochastic) trend, seasonal, regression elements, and disturbance terms, all of which can be independently modeled, allowing for the study of their evolution through time and a clear understanding of their contribution to the final predictions [79].

A generalized expression of the decomposition of a time series is given by Equation (8), such that:

$$y_t = \mu_t + \psi_t + \gamma_t + \varepsilon_t, t = 1, \ldots, T \tag{8}$$

where $\mu_t$ is the trend, $\psi$ is the cycle, $\gamma_t$ is the seasonal, and $\varepsilon_t$ is the irregular white noise component. All of the four components are stochastic and are driven by mutually uncorrelated disturbances [80]. The framework developed in this study to predict e-commerce retail sales embeds the "StructTS" function in R's "stats" package that uses maximum likelihood to determine the model's parameters.

Artificial neural networks (ANNs) are inspired by biological systems, particularly by the way biological nervous systems process information [42]). ANNs have been proven capable of modeling complex real-world systems by fully taking nonlinearities into account [81]. The basic elements that describe an ANN are the nodes or the basic processing elements of the network, the network architecture reflecting the connections between nodes; and the training algorithm used to find values of the network parameters for performing a particular task [82]. The type of ANN where each layer of nodes receives inputs from the previous layer is called multi-layered feed-forward network or multilayer perceptron (MLP). The models are referred to as feed-forward since there are no feedback connections so that predictions issued by the model (i.e., outputs) are not fed back into itself [83]. Many different ANN models have been proposed in the literature, depending on the nature of the task assigned to the network, as there are also numerous variations in modeling the nodes [84]. Lagged values from time series data are frequently utilized as inputs in an ANN structure, which is thus defined as neural network autoregression (NNAR) when applied to time series data [85]. NNAR models can be defined as NNAR *(p,P,k)m* for seasonal data, where *m* is the seasonal period, *p* denotes non-seasonally lagged inputs for the linear AR process, *P* are seasonal lags for the AR process, and *k* denotes the number of hidden layer nodes.

The neural network autoregression model can be written as:

$$Y = f(H) = f(W * X + B), X = [y(t-1), y(t-2), \ldots, y(t-p)] \tag{9}$$

where *Y* is the output vector of predicted values, f is the activation function, *H* is the vector of n nodes in the hidden layer, *W* represents the weight matrix between the input and the hidden layers, *X* is the vector of inputs containing the lagged values of the observed data and B is a bias vector [86]. Hence, the first layer receives the lagged values of the time series representing first e-commerce retail sales and then the share of retail as inputs, then, the weighted inputs are combined linearly and fed-forward to the hidden layer where the activation function is applied. The result is finally sent to the output layer of the network. The integrated framework in this study automatically fits multi-layer feed-forward neural networks with a single hidden layer, k nodes, and p-lagged inputs using the "nnetar" function in the R "forecast" package. To automatically determine the parameters p and P,

the method performs 25 iterations while minimizing AIC. Additionally, the hidden layer's node count is estimated is $k = (p + P + 1)/2$.

It should be mentioned that several criteria must be met for robust time series forecasting, including employing appropriate test and training sets, performing ex-ante validations, comparing projections to established models, and cross-validation ([74,87,88]). First, this study accounts for cross-validation by analyzing two training samples, in path A and path B, respectively. Second, the training and testing windows have adequate lengths, whereas the use of out-of-sample testing windows meets the ex-ante validation criterion. Finally, the comparative perspective is adopted by evaluating the forecasting ability of six statistical and machine-learning models, including well-established time-series models, among which NNAR and TBATS can handle nonlinear features in time series.

### 3.5. Forecast Error Measures

A model's predictive accuracy is determined by estimating its forecast errors by comparing the actual and predicted values. Thus, the forecast error equals:

$$e_{T+h} = y_{T+h} - \hat{y}_{T+h|T} \tag{10}$$

with $\{y1, \ldots, yT\}$ denoting the training set data and $\{yT + 1, yT + 2, \ldots\}$ the test set data.

For the purpose of analyzing the prediction accuracy of the model, in this study we estimate the following forecasting accuracy metrics: mean absolute error (MAE), root mean squared error (RMSE), mean absolute percentage error (MAPE), mean absolute scaled error (MASE), Theil's U statistic, and Lag 1 autocorrelation of error (ACF1). Estimation equations for all measures are given in Appendix A.2, whereas further details on these accuracy measures and their interpretation are provided by [74,89].

### 3.6. Further Robustness Checks: Tests for Superior Forecasting Performance

The Diebold and Mariano test (DM) is estimated to compare model performance at each forecasting horizon employed in the empirical investigations. In order to assess the null hypothesis of equal prediction performance between two competing models, the test developed by [90] and its extended version by [91] has been broadly applied in the literature [92,93]. The DM test statistic is applicable to forecast errors that are contemporaneously correlated, serially correlated, and nonnormal [94]. It is assumed that the loss connected to forecast $i$ is a function of the forecast error, $e_{it}$, indicated by $g(e_{it})$, which is typically the square or the absolute value of $e_{it}$. The loss difference between the two forecasts is further assessed to be:

$$d_t = g(e_{1t}) - g(e_{2t}) \tag{11}$$

and the following is the null hypothesis to be evaluated:

$$HO : E(d_t) = 0, \forall t \tag{12}$$

Overall, despite contesters, DM test remains useful for informing on the comparative historical predictive performance [95].

In this research, the integrated framework estimates the DM test via the "dm.test" function included in the "forecast" package of the R environment.

## 4. Results

The accuracy metrics for the six statistical and machine-learning models' out-of-sample predicting performance embedded into the integrated framework and employed to predict the evolution of the two e-sales variables are reported in Tables 1 and 2. Results show that the exponential smoothing state space model with Box-Cox transformation, ARMA errors, trend, and seasonal components (TBATS) model can provide the best forecast for e-sales at a horizon of seven quarters, while both the TBATS and STS models out-perform

when forecasting e-share over the April 2017–December 2019 period. Note that all models exhibit sub-unitary Theil's U statistics. in path A except for NNAR, meaning that all given predictions outperform the naive technique in path A of the algorithm. This is explained by the constantly increasing trend in the two variables over the testing period that is missed by the naïve forecast, which always predicts a flat line. For path B, accuracy measures are significantly higher than for path A, indicating that none of the models has captured the turbulent evolution of the e-commerce series. However, the TBATS and the STS model register similar performances outside the sample for both series in the second scenario (path B).

Of note, the DM test of equal forecast accuracy between two competing models (i.e., the best and second-best performers on the alternate test data) confirms that TBATS is over-performing in predicting e-commerce retail sales in both paths A and B within the array of six predictive models. For the e-share series, the test of superior forecasting ability lacks statistical significance in path A and thus indicates that there is no clear distinction between the predictive ability of the two best performers (i.e., STS and TBATS), whereas in path B the superior forecasting ability of TBATS is again confirmed. Figures 5 and 6 contain scatterplots showing predicted vs. actual values for the two e-commerce series given by the over-performing models (i.e., TBATS) and, for comparative purposes, by the neural network models. All charts indicate that TBATS has been able to forecast the evolution of both series fairly well (adjusted $R^2$ = 97.62% and RMSE = 1317.1 for e-sales and adjusted $R^2$ = 98.34% and RMSE = 0.0634 for e-share), whereas NNAR often underestimates its projections (adjusted $R^2$ = 94.63% and RMSE = 1990.8 for e-sales and adjusted $R^2$ = 96.85% and RMSE = 0.0883 for e-share). In all plots, the regression line between predicted vs. actual values, as well as the perfect-fit line are also drawn, attesting to the superior forecasting ability of TBATS. Moreover, Appendix A.1 also reflects the excellent out-of-sample forecasting ability of TBATS over the pre-pandemic testing period (path A) for the two e-commerce time series.

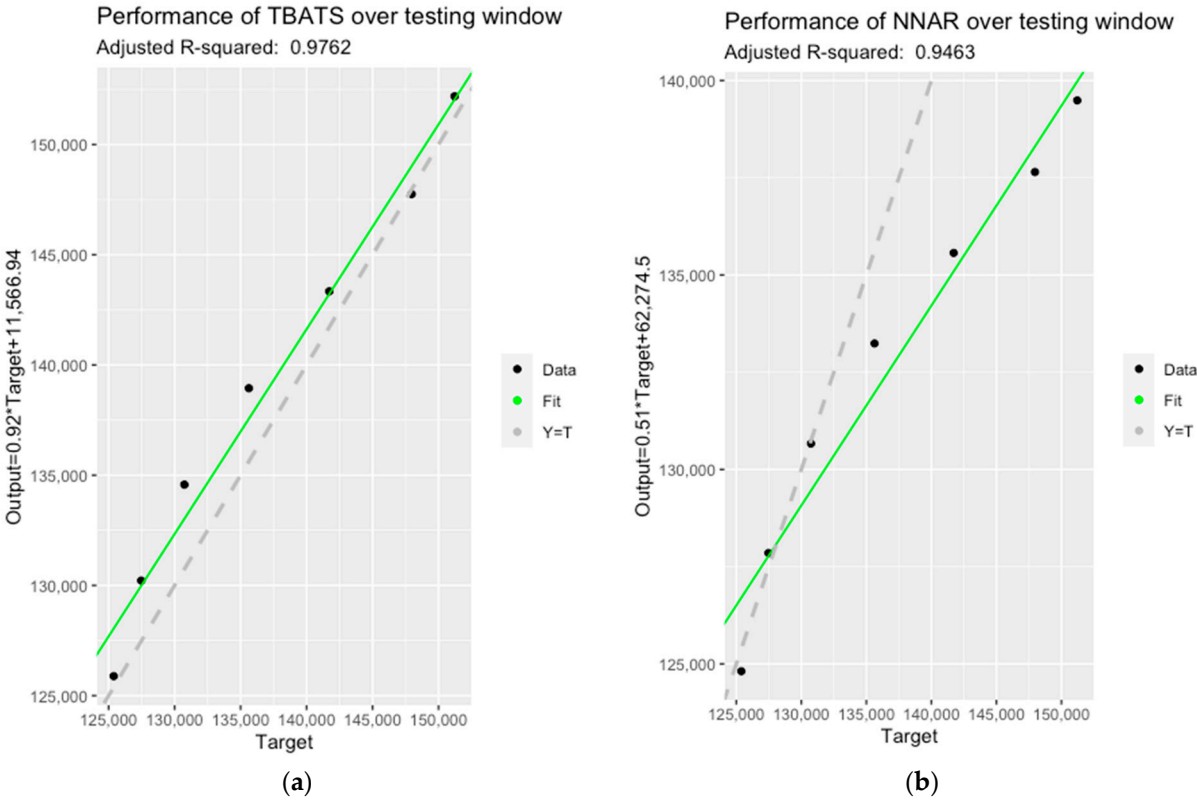

(**a**)  (**b**)

**Figure 5.** Model performance over testing window (e-commerce retail sales, path A): TBATS (panel **a**), and NNAR (panel **b**). Source: estimation results.

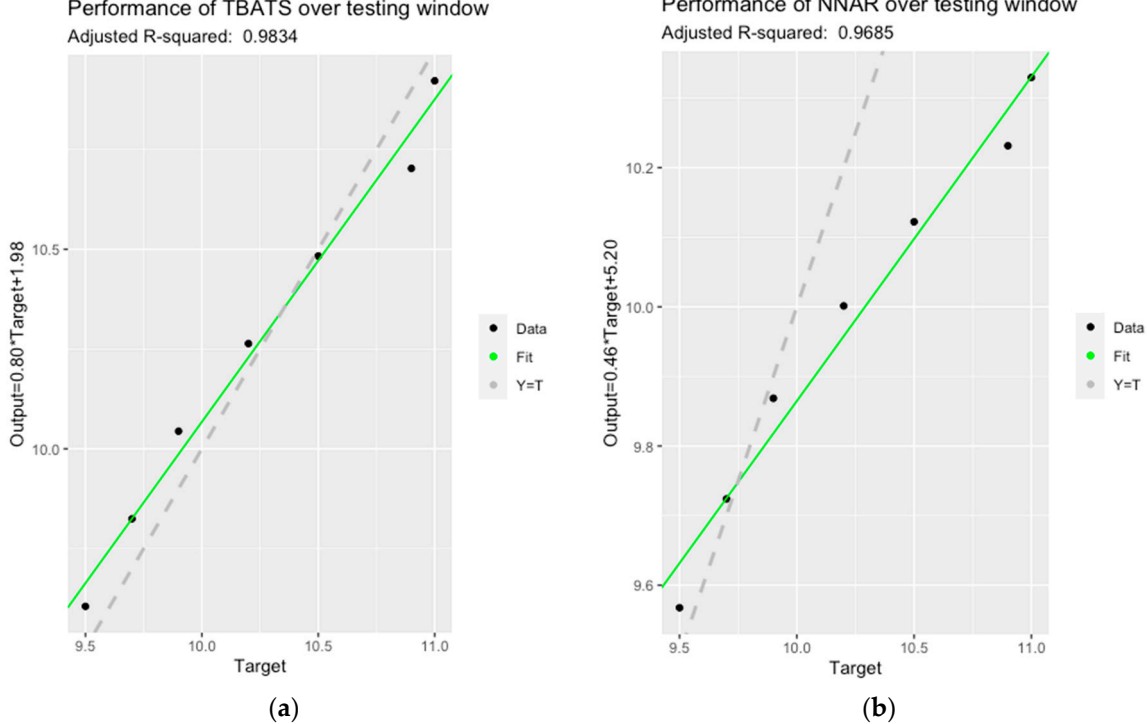

**Figure 6.** Model performance over testing window (e-commerce retail share, path A): TBATS (panel **a**), and NNAR (panel **b**). Source: estimation results.

Results indicate that the difference between all accuracy metrics for the TBATS models and the neural network auto-regression model (NNAR) model is very high for both series and all test-sets, with the TBATS model significantly outperforming the neural network in every case.

TBATS therefore emerges as the overall best option when choosing to rely on a single predictive model for e-sales and e-share at the chosen horizon. However, NNAR is unable to offer competitive predictions for the evolution of the two e-commerce series (i.e., e-sales and e-share) across different test periods.

**Table 1.** Performance metrics for the six models on ECOMSA (path A and path B).

| Path A: Training Set (October 1999 to March 2017) and Test Set (April 2017 to December 2019) | | | | | | |
|---|---|---|---|---|---|---|
| **Predictive Model** | **MAE** | **MAPE** | **MASE** | **RMSE** | **Theil's U** | **ACF1** |
| NNAR | 4662.79 | 3.19 | 2.74 | 6592.65 | 1.48 | 0.64 |
| ETS | 2131.34 | 1.59 | 1.25 | 2548.14 | 0.63 | 0.36 |
| ARIMA | 1923.24 | 1.44 | 1.13 | 2353.49 | 0.58 | 0.41 |
| STS | 1978.93 | 1.48 | 1.16 | 2421.46 | 0.60 | 0.39 |
| HW | 2102.57 | 1.57 | 1.24 | 2526.42 | 0.63 | 0.37 |
| TBATS | 1885.01 | 1.41 | 1.11 | 2301.64 | 0.57 | 0.40 |
| DM (TBATS/ARIMA) (*p*-value) | 0.0357 ** | | | | | |
| **Path B: Training Set (October 1999 to June 2017) and Test Set (July 2017 to September 2021)** | | | | | | |
| **Predictive Model** | **MAE** | **MAPE** | **MASE** | **RMSE** | **Theil's U** | **ACF1** |
| NNAR | 32,950.70 | 26.69 | 24.68 | 33,666.45 | 1.74 | 0.50 |
| ETS | 15,516.73 | 12.60 | 11.99 | 15,846.25 | 1.47 | 0.41 |
| ARIMA | 15,530.94 | 12.11 | 11.56 | 15,860.50 | 1.52 | 0.41 |
| STS | 15,540.40 | 12.59 | 11.58 | 15,870.03 | 1.47 | 0.41 |
| HW | 15,530.95 | 13.67 | 11.67 | 15,860.51 | 1.48 | 0.41 |
| TBATS | 14,702.29 | 11.59 | 11.01 | 15,000.50 | 1.47 | 0.39 |
| DM (TBATS/STS) (*p*-value) | 0.0729 ** | | | | | |

** significant at 5%.

**Table 2.** Performance metrics for the six models on ECOMPCTSA (path A and path B).

| Path A: Training Set (October 1999 to March 2017) and Test Set (April 2017 to December 2019) | | | | | | |
|---|---|---|---|---|---|---|
| **Predictive Model** | **MAE** | **MAPE** | **MASE** | **RMSE** | **Theil's U** | **ACF1** |
| NNAR | 0.28 | 2.60 | 2.31 | 0.38 | 1.45 | 0.61 |
| ETS | 0.11 | 1.05 | 0.89 | 0.12 | 0.46 | 0.29 |
| ARIMA | 0.11 | 1.08 | 0.91 | 0.12 | 0.47 | 0.31 |
| STS | 0.10 | 1.01 | 0.85 | 0.12 | 0.45 | 0.28 |
| HW | 0.12 | 1.15 | 0.96 | 0.13 | 0.49 | 0.34 |
| TBATS | 0.10 | 1.02 | 0.86 | 0.12 | 0.45 | 0.26 |
| DM (STS/TBATS) (*p*-value) | 0.653 | | | | | |
| **Path B: Training Set (October 1999 to December 2019) and Test Set (January 2020 to October 2021)** | | | | | | |
| **Predictive Model** | **MAE** | **MAPE** | **MASE** | **RMSE** | **Theil's U** | **ACF1** |
| NNAR | 1.72 | 13.65 | 14.81 | 2.26 | 1.21 | 0.72 |
| ETS | 0.91 | 6.73 | 7.80 | 1.53 | 1.08 | 0.56 |
| ARIMA | 0.91 | 6.61 | 7.08 | 1.52 | 1.09 | 0.55 |
| STS | 0.87 | 6.34 | 7.61 | 1.52 | 0.87 | 0.55 |
| HW | 0.90 | 6.56 | 7.68 | 1.50 | 1.08 | 0.55 |
| TBATS | 0.76 | 5.51 | 6.60 | 1.38 | 0.87 | 0.52 |
| DM (STS/TBATS) (*p*-value) | 0.0217 ** | | | | | |

** significant at 5%.

We next produce the expected e-sales and e-share values for the next 7 quarters corresponding to the pandemic period (from January 2020 to October 2021) in the absence of the COVID-19 pandemic (or during normal conditions). For this task, the best-performing models over the test period are selected. Consequently, point estimates for e-sales and e-share over the seven quarters that correspond to the pandemic period are issued through the TBATS model. Finally, abnormal values for the two variables are estimated by Equations (1) and (2) and reported. Table 3 centralizes all results for the e-sales variable, whereas the corresponding results for the e-share series are reported in Table 4.

**Table 3.** Excess e-commerce retail sales over the pandemic window (in millions of dollars).

| | **Real Value (mil. USD)** | **Estimated Value (mil. USD)** | **Excess E-Sales (mil. USD)** |
|---|---|---|---|
| 2020 Q1 | 154,575 | 155,453.7 | −878.7 |
| 2020 Q2 | 203,847 | 159,944.5 | 43,902.5 |
| 2020 Q3 | 201,385 | 164,464.1 | 36,920.9 |
| 2020 Q4 | 199,665 | 169,012.3 | 30,652.7 |
| 2021 Q1 | 215,290 | 173,588.4 | 41,701.6 |
| 2021 Q2 | 221,951 | 178,192.2 | 43,758.8 |
| 2021 Q3 | 214,586 | 182,823.2 | 31,762.8 |
| Sum | | | 227,820.6 |

**Table 4.** Excess e-commerce retail sales as a percent of total sales over the pandemic window (%).

| | **Real Value(%)** | **Estimated Value (%)** | **Excess E-Share (%)** |
|---|---|---|---|
| 2020 Q1 | 11.4 | 11.25 | 0.15 |
| 2020 Q2 | 15.7 | 11.49 | 4.21 |
| 2020 Q3 | 13.8 | 11.73 | 2.07 |
| 2020 Q4 | 13.6 | 11.97 | 1.63 |
| 2021 Q1 | 13.6 | 12.21 | 1.39 |
| 2021 Q2 | 13.3 | 12.45 | 0.85 |
| 2021 Q3 | 13.0 | 12.69 | 0.31 |
| Sum | | | 10.61 |

Of note, estimations corresponding to the pandemic period indicate a continuation of the increasing trends both for e-commerce retail sales as well as for its penetration rate (i.e., the share of e-commerce in total retail sales). Hence, while the COVID-19 has significantly accelerated the rhythm of growth of the e-commerce sector, it does not necessarily mean that this increase was entirely caused by the pandemic.

We report an excess of e-commerce retail sales of $43.9 billion in 2020 Q2, and an excess of 4.2 percent for the share of e-commerce in total retail sales for the same time frame. As can be seen, in this first wave the pandemic spurred e-commerce and induced high levels of excess e-sales and e-share, while in the second year of the pandemic e-commerce retail sales registered almost similar excess levels, while the share of e-commerce saw much more modest increases as compared to the first wave of the pandemic. This further implies that the structural transformation of the retail sales sector caused by the Black-Swan event came to a halt after the first half of 2020.

In the last stage of this investigation, the best-performing predictive model for each e-commerce series in terms of out-of-sample forecasting ability is fitted to the entire series and further employed to produce point forecasts for the 2021 Q4–2025 Q4 period (i.e., with a forecast horizon h = 17).

Table 5 contains the estimated values for the two series. Results indicate that the current increasing trend will continue through the end of 2025.

**Table 5.** Trend estimation from 2021Q4 to 2025Q4 for e-sales (ECOMSA) and e-share (ECOMPTCSA).

|  | Point Forecast E-Sales (mil. USD) | Point Forecast E-Share (%) |
|---|---|---|
| 2021 Q4 | 225,330.1 | 13.47 |
| 2022 Q1 | 233,242.8 | 13.67 |
| 2022 Q2 | 241,361.6 | 13.87 |
| 2022 Q3 | 249,690.0 | 14.08 |
| 2022 Q4 | 258,231.6 | 14.28 |
| 2023 Q1 | 266,989.9 | 14.49 |
| 2023 Q2 | 275,968.5 | 14.65 |
| 2023 Q3 | 285,171.2 | 14.89 |
| 2023 Q4 | 294,601.5 | 15.09 |
| 2024 Q1 | 304,263.2 | 15.30 |
| 2024 Q2 | 314,160.0 | 15.56 |
| 2024 Q3 | 324,295.7 | 15.71 |
| 2024 Q4 | 334,674.1 | 15.93 |
| 2025 Q1 | 345,298.9 | 16.11 |
| 2025 Q2 | 356,174.2 | 16.32 |
| 2025 Q3 | 367,303.6 | 16.54 |
| 2025 Q4 | 378,691.2 | 16.72 |

## 5. Discussion

The integrated framework developed in this study is employed for two alternative empirical investigations. As such, the algorithm consecutively runs on two distinct paths. In path A, the algorithm is instructed to train all models on the pre-pandemic window and produce forecasts over the test data corresponding to the seven quarters of the pandemic period, spanning from 2020 Q1 to 2021 Q3. Predicted values are used as proxies for the normal sector evolution in the absence of the pandemic. Then, excess indicators are estimated to offer a quantitative assessment of the COVID-19 impact on the US e-commerce sector. The results indicate abnormal e-commerce retail sales totaling $227.820 billion and a cumulative excess increase of 10.61 percent in the share of e-commerce in retail sales over the pandemic period spanning from January 2020 to September 2021 Q3. Another significant finding is that the growing tendency would have continued even if the black swan event had not occurred, albeit the pandemic has significantly accelerated its evolution.

Subsequently, in path B, the algorithm uses the most recent test data to issue forecasts of the two series for the next five years. Estimations imply that the upward trend will

continue both in size and weight of the e-commerce sector. Total e-commerce retail sales are thus predicted to grow by 76.47% over the November 2021-December 2025 period and to thus reach $378.691 billion, accounting for an estimated 16.72 percent of US retail sales by the end of 2025. This in turn corresponds to a further 3.72% increase in the share of e-commerce in total retail sales over the forecasting horizon, which confirms the definitive change in consumer preferences.

### 5.1. E-Commerce Trends and the COVID-19 Impact

Overall, research findings reinforce previous results in the literature, showing that e-commerce, enabled by computer and internet technologies, has registered an increasing trend over the past decades [96] and has surged following the outbreak of the COVID-19 pandemic ([13,18,19,56]). However, it also confirms that the pandemic did not cause, albeit it has strongly accelerated, a steadily increasing trend for both e-commerce variables [97]. Furthermore, this study brings further evidence that the structure of the e-commerce sector in the United States has been constantly evolving in the past decade, as [98] acknowledges, with non-store retailers driving the growth of e-commerce retail sales and the increase in e-commerce penetration rate.

In quantitative terms, research findings assess the impact of the COVID-19 pandemic as $227.820 billion excess sales over the pandemic period spanning from 2020 Q1 to 2021 Q3 and a cumulative excess increase of 10.61 percent in the share of e-commerce in retail sales as compared to expected values in the absence of the black swan event. While a continuation of the increasing trend was expected, none of the models could accurately capture the strong trends induced by the pandemic and most of the excess gains in the e-commerce variables are concentrated in the second quarter of 2020. These estimates thus support prior findings that the pandemic-induced health and economic crisis has accelerated the usage of digital channels, particularly e-commerce [6–9].

Results also show that the increasing trend is expected to continue by the end of the forecasting horizon (i.e., 2025 Q4). Thus, we forecast US retail e-commerce sales to grow 76.47% over the November 2021-December 2025 period, reaching $378.691 billion by the end of 2025. E-commerce share in total retail sales is expected to increase by 3.72%, as e-commerce will account for an estimated 16.72 percent of US retail sales by the end of the forecasting horizon.

This brings further proof to confirm an aforementioned finding that the growth of e-commerce is not a result of the COVID-19 crisis (although it is the case that the pandemic has accelerated it), resulting in a shift in consumer behavior and spurring e-commerce retail sales. Official sector-level statistics on the level of the e-sales during the pandemic are not yet available, but we expect previous distinct trends across sectors to have continued, and even to have deepened. Our findings also agree with [13,22], showing that after B2C e-commerce surged during the pandemic, this trend is expected to last in the aftermath of the pandemic owing to the permanent shift of retail sales online in the vast majority of cases. Our projections however surpass Statista's predictions cited by [99], which expect the North American e-commerce sector to grow by 35% over the next five years.

### 5.2. Characteristics of E-Commerce Time Series

Moreover, this work adds to the evidence presented by [25], while also demonstrating that e-commerce retail sales time series are characterized by trend and seasonal components that should be accounted for in producing accurate forecasts. The study demonstrates that a TBATS model, when properly calibrated, can accurately forecast the evolution of the two e-commerce series. The ability of the TBATS model to adapt to the historical data results in better forecasting performance on alternative test data windows (i.e., paths A and B of the algorithm that is run within the integrated platform) compared to other competing models (both linear and non-linear) employed in this research (i.e., ETS, HW, ARIMA, STS, and NNAR). Estimations find that TBATS is over-performing in forecasting both e-commerce time-series and across different testing set data, confirming its superior ability to handle

typical nonlinear data features, while considering any autocorrelation in the residuals and multiple seasonal components [76]. In general, we thus conclude that TBATS models can forecast the trend movement and seasonal fluctuations fairly well, and back [100]) and [101] by showing that individual nonlinear models perform better than linear models in forecasting aggregate retail sales. This is also in line with the results of [67] that arrives at the same conclusion in estimating the impact of the pandemic on a time series that has registered a similar evolution after January 2020 (i.e., excess hospitalizations caused by the COVID-19 pandemic). However, these findings deviate from [51] that identify the neural network model as best for aggregate retail sales forecasting, whereas trigonometric models are not found to be useful for the task in this prior study. Our divergent results that rank NNAR as one of the worst-performing models for forecasting e-commerce series and thus unable to capture seasonal or trend variations effectively could be explained by the fact that we did not pre-process the data through de-trending or de-seasonalization, which has been shown to dramatically reduce forecasting errors of neural network models [102].

### 5.3. Policy Implications

In light of these projections and given previous studies that acknowledge e-commerce as a pollution-mitigating factor [32] and driver of the circular economy [10], we argue that policies that encourage digital transformation and innovation, particularly for small and medium-sized businesses, should be prioritized. We thus also support the conclusion of [103] and agree that digital transformation should constitute a top management priority.

Hence, future predictions for the full path of the series of e-commerce retail sales and for the share of e-commerce in retail sales up to the end of 2025 that are issued through the integrated platform are paramount not only in decision-making processes at the company level ([24,25,33,104,105]) but also for policymakers that require forecasts at different horizons to make informed decisions [106].

### 6. Conclusions

Accurate retail sales forecasting is paramount for retail business management, influencing a range of operational decisions, as businesses must continually adapt to changing circumstances in today's competitive business environment. Consequently, producing accurate forecasts of retail sales has become equivalent to acquiring or maintaining business success.

The COVID-19 pandemic has significantly impacted the retail sales sector, expanding the scope of e-commerce and accelerating its expansion. Consequently, as retail sales have plummeted following the pandemic outbreak, e-commerce has spurred. The increasing trend of the e-commerce sector is expected to continue, as some of its changing circumstances will most likely be long-term, given the potential for new pandemic waves, the efficiency of the new consumer behavior, learning expenses, and the urge for entrepreneurs to rip the benefits of investing in digital retail channels [13].

Furthermore, it has long been recognized that retail sales are an established leading indicator for important macro-economic variables, whereas e-commerce is also a mitigating factor for polluting emissions and a driver for the circular economy. Consequently, retail sales and in particular e-commerce are significantly influencing policymaking processes.

Hence, owing to the aforementioned factors, interest has widely shifted from forecasting retail sales toward forecasting e-commerce sector indicators.

However, retail sales time series (including e-commerce) often exhibit strong trend and seasonal variations presenting challenges in developing effective forecasting models. Consequently, previous studies disagree on the best model for this task, and accurately modeling and forecasting retail sales series remains an outstanding research topic. Considering the stronger trends and fluctuations encountered in e-commerce series as compared to retail sales, the task of accurately capturing the specificities of e-commerce variables and producing accurate forecasts is that much more challenging.

This paper contributes to e-commerce research by suggesting an integrated approach to model and predict its evolution. The main goal of this study is thus to develop an integrated framework to forecast two e-commerce time series: e-commerce retail sales and the share of e-commerce in total retail sales. The framework embeds an algorithm capable of efficiently estimating six statistical and machine-learning methods (i.e., ETS, HW, ARIMA, STS, TBATS, and NNAR). The algorithm automatically searches for the best in-sample and out-of-sample model parameters. As such, it applies various fitness functions (AIC, AICc, ML) to all model specifications that pass the diagnostic checking on the training window to identify the best in-sample parameters for each of the six models. Then, the optimal six model specifications selected in-sample through the specific fitness function are employed for forecasting on the out-of-sample window. Multiple model accuracy measures, both scaled and scale-free (i.e., MAE, MAPE, MASE, RMSE, Theil's U, ACF1) are subsequently estimated to assess the predictive ability of the six best (in-sample) model specifications. Finally, the optimal (out-of-sample) forecasting model for each series and each testing window automatically emerges as the one that minimizes forecasts errors. Results indicate that the overall out-of-sample forecasting performance of the TBATS model is best for all circumstances.

As far as we know, this is the first time that this approach is proposed for retail sales forecasting (including e-commerce), whereas a quantitative assessment of the COVID-19 impact on the e-commerce sector through the estimation of its excess (or abnormal) evolution constitutes another novelty. Overall, we argue that the framework proposed in this paper can be a useful managerial tool in decision making, owing to its efficiency, flexibility, and prediction accuracy compared to the state-of-the-art approaches. Results of this research further indicate that on one hand governments should prioritize policies that encourage digital transformation and innovation, and support business adaptation, particularly for small and medium-sized enterprises whereas, on the other hand, digital transformation should constitute a top management priority. The integrated framework capable to produce forecasts of retail sales time series, and assessing abnormal evolutions, or the magnitude of the impact of black swan events, is thus applicable both for retailers and policymakers.

**Funding:** This research received no external funding.

**Data Availability Statement:** Data employed in the empirical investigations are publicly available on the Federal Reserve Bank of St. Louis (FRED) website.

**Conflicts of Interest:** The authors declare no conflict of interest.

## Appendix A
*Appendix A.1*

Figure A1 reflects the forecasted trend produced by the over-performing predictive model (i.e., TBATS) for e-commerce retail sales (panel a) and e-commerce retail share (panel b) over the testing window for path A of the algorithm.

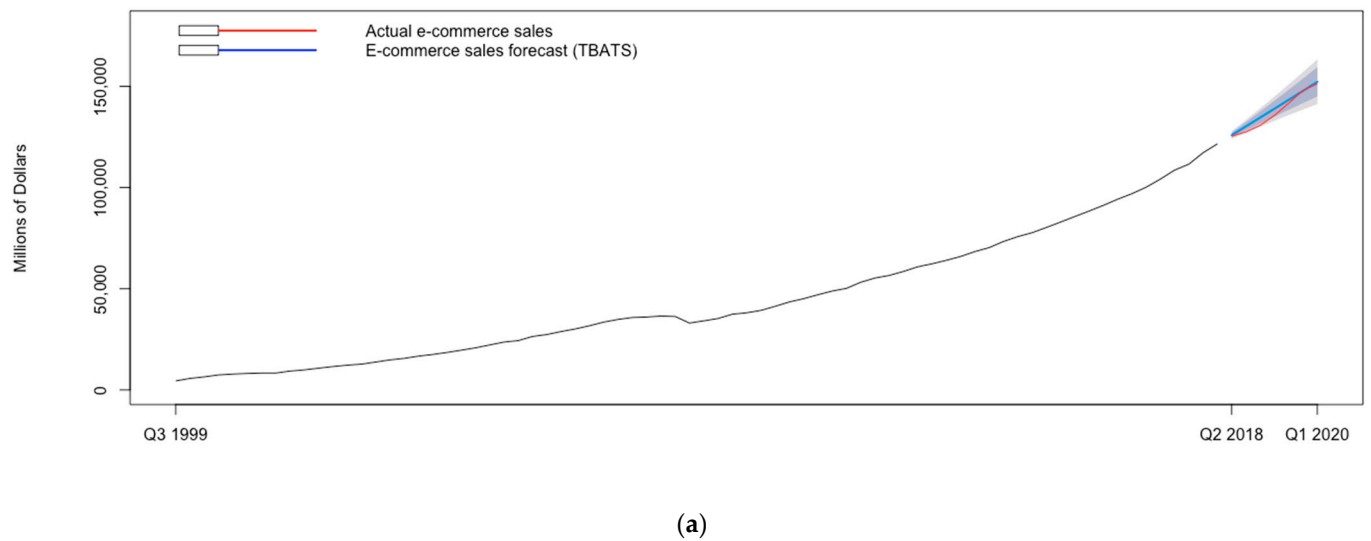

(**a**)

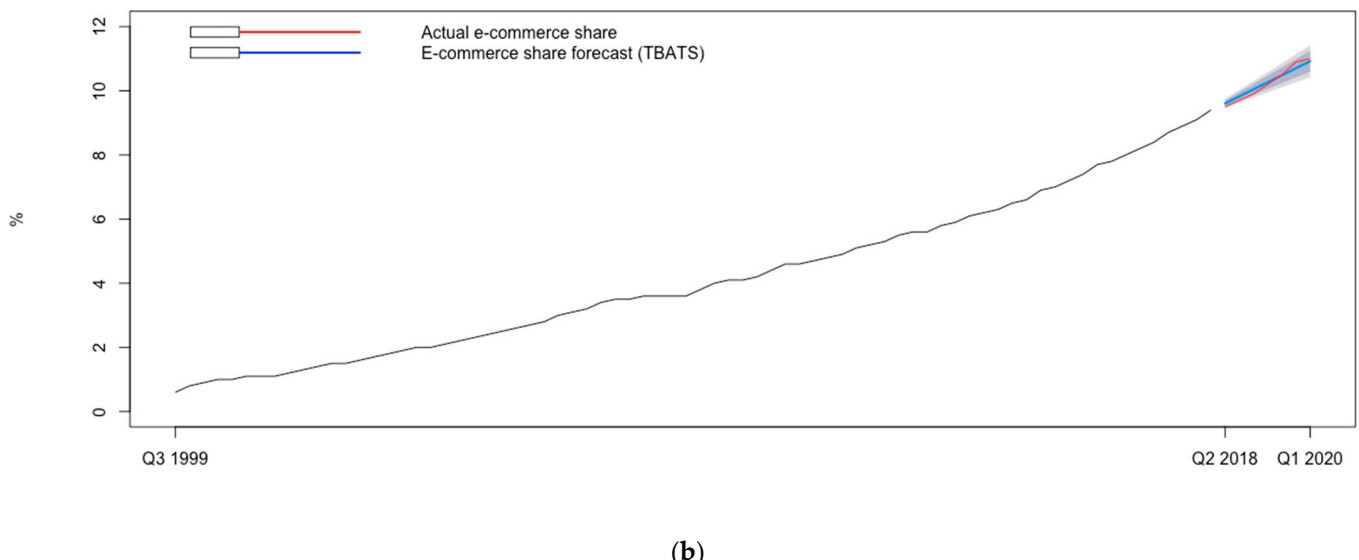

(**b**)

**Figure A1.** (**a**) Forecasted trend for e-commerce retail sales over April 2018–January 2020 issued with TBATS. Source: estimation results. (**b**) Forecasted trend for e-commerce share in total retail sales over April 2018–January 2020 issued with TBATS. Source: estimation results. Source: estimation results.

*Appendix A.2. Accuracy Measures*

Mean absolute error:

$$MAE = \sqrt{\frac{1}{N}\sum_{i=1}^{N}|y_i - \hat{y}_l|} \tag{A1}$$

Root mean squared error:

$$RMSE = \sqrt{\frac{1}{N}\sum_{i=1}^{N}(y_i - \hat{y}_l)^2} \tag{A2}$$

Mean absolute percentage error:

$$MAPE = mean(|p_t|) \tag{A3}$$

where: $p_t = \frac{100e_t}{y_t}$.

Mean absolute scaled error:

$$\text{MASE} = \text{mean}(|q_j|) \tag{A4}$$

where $q_j$ is independent of the scale of the data and is defined as: $q_j = \dfrac{e_t}{\frac{1}{N-1}\sum\limits_{i=2}^{N}|y_t-y_{t-1}|}$ for

non-seasonal series and as: $q_j = \dfrac{e_t}{\frac{1}{N-m}\sum\limits_{i=m+1}^{N}|y_t-y_{t-m}|}$ for seasonal time series.

Theil's U statistic:

$$Theil's U = \frac{RMSE_1}{RMSE_2} \tag{A5}$$

where $RMSE_2$ is the RMSE of the naive technique, which always produces a flat line, and $RMSE_1$ is the RMSE metric obtained for the main model; Theil's U will take values between o and 1, with zero indicating a perfect prediction [107].

Lag 1 Autocorrelation of Error:

$$ACF1 = \frac{\hat{\gamma}(1)}{\hat{\gamma}(0)} \tag{A6}$$

where $\gamma$ is the sample autocovariance function.

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
