# Peer review of "Integrated Framework to Assess the Extent of the Pandemic Impact on the Size and Structure of the E-Commerce Retail Sales Sector and Forecast Retail Trade E-Commerce"

_electronics, doi:10.3390/electronics11193194_

Round 1

Reviewer 1 Report

1- Novelty of the research is not clear 

2- Finally, I suggest the authors improve the figures as they look stretched. 

3.Which software package have they used to build the architecture? Python or R? Please share your model and data via GitHub.

4- All  equation is arrange  and numbering properly 

5-  I suggested author used  artificial intelligence models  to compare with results of  tradition approaches that he used

6-  The dataset is non-linear   data, how you get high accuracy by using linear time series models.

7- Where is  results of MSE and RMSE errors, it is not there?

8- I suggest authors  to draw regression plot for investigate  between the predicting value and observation values 

9- comparison between your results and existing results is not find

Reviewer 2 Report

The paper is high quality and very well written.

Authors may suggest reducing the introduction before research questions and contributions a lit bit to avoid readers from getting distracted.

I would suggest authors add a few recent references, such as 

https://doi.org/10.1186/s12859-022-04751-6

I would suggest authors write the steps of the algorithm in a well-defined text box for the algorithm and label it.

I would also suggest the author align the equations properly, currently, they are larger in font size and not aligning.

The evaluation metrics are not properly placed and are mixed. authors can arrange them in order and place for ease of readability.

In the results, why the authors' model has MAE too low (which is appreciated) but compared to other models, authors need to explain.

The discussion section and conclusions are way too long, authors should move a lot of conclusion parts into the discussion and preferably give a heading to each para in the discussion for ease of readability.

Reviewer 3 Report

This paper contributes to e-commerce research by suggesting an integrated approach to model and predict its evolution. The main goal of this study is thus to develop an integrated framework to forecast two e-commerce time series: e-commerce retail sales and the share of e-commerce in total retail sales. The framework embeds an algorithm capable of efficiently estimating six statistical and machine-learning methods (i.e. ETS, HW, ARIMA, STS, TBATS, and NNAR). The algorithm automatically searches for the best in-sample and out-of-sample model parameters. As such, it applies various fitness functions (AIC, AICc, ML) to all model specifications that pass the diagnostic checking on the training window to identify the best in-sample parameters for each of the six models. Then, the optimal six model specifications selected in-sample through the specific fitness function are employed for forecasting on the out-of-sample window. Several scaled and scale-free model accuracy metrics (i.e. MAE, MAPE, MASE, RMSE, Theil’s U, ACF1) are subsequently estimated to assess the predictive ability of the six best (in-sample) model specifications. Finally, the optimal (out-of-sample) forecasting model for each series and each testing window automatically emerges as the one that minimizes forecasts errors. Results indicate that the overall out-of-sample forecasting performance of the TBATS model is best for all circumstances. The study thus demonstrates that a TBATS model, when properly calibrated, can accurately forecast the evolution of the two e-commerce series. The ability of the TBATS model to adapt to the historical data results in better forecasting performance on alternative test data windows (i.e. paths A and B of the algorithm that is run within the integrated platform) compared to other competing models (both linear and non-linear) employed in this research (i.e. ETS, HW, ARIMA, STS, and NNAR).

 ***

The paper is very badly formatted. There are some confusing symbols in the figures and formulas.

The author argues that the integrated framework capable of producing time-series forecasts of retail sales and assessing unusual changes or the magnitude of the impact of black swan events is applicable to both retailers and policy makers, but there is no example of such an application.

Round 2

Reviewer 1 Report

Thanks for making some chnages but they are not satifying and acceptable

1- Regression plot is wrong, you have to draw regtression plot  for invetigate between prediction value and actual values 

2.  Authors  can use AI model for comapriong between the tradition model and propsoed model

3.comparison between your results and existing results is not find  in table I want to the prediction errors of your model and exsting models by using  MSE or RMSE

Round 3

Reviewer 1 Report

Regression plot still not clear 

plz look to this article  how they draw regression plot 
